# Differences in Breast Cancer Subtypes among Racial/Ethnic Groups

**DOI:** 10.3390/cancers16203462

**Published:** 2024-10-12

**Authors:** Tamlyn Sasaki, Akash Liyanage, Surbhi Bansil, Anthony Silva, Ian Pagano, Elena Y. Hidalgo, Corinne Jones, Naoto T. Ueno, Yoko Takahashi, Jami Fukui

**Affiliations:** 1John A. Burns School of Medicine, University of Hawai’i, Honolulu, HI 96813, USA; tamlyn21@hawaii.edu (T.S.); nueno@cc.hawaii.edu (N.T.U.); 2Translational and Clinical Research Program, University of Hawai’i Cancer Center, Honolulu, HI 96813, USA; ipagano@cc.hawaii.edu (I.P.); ytakahashi@cc.hawaii.edu (Y.T.); 3Queen’s Medical Center Oncology Data Registry, Honolulu, HI 96813, USA; 4Kapi’olani Medical Center for Women and Children, Honolulu, HI 96822, USA; corinne.jones@kapiolani.org; 5The Queen’s Health Systems, Honolulu, HI 96813, USA

**Keywords:** breast cancer, race, Hawaii, subtype, disparities

## Abstract

**Simple Summary:**

Breast cancer is a complex disease with several subtypes that impact different populations in various ways. This study focuses on the diverse ethnic population of Hawai’i. We aim to investigate whether there are differences in breast cancer subtypes among various racial and ethnic groups that could contribute to disparities in breast cancer outcomes. This study analyzes the incidence and prevalence of breast cancer subtypes in these groups, considering factors such as age and tumor biology. By identifying subtype-specific risks and outcomes, we hope to provide insights that could lead to more tailored and effective treatments, thereby improving prognosis and reducing mortality disparities in diverse communities. Our findings have important implications for clinical practice in regions with diverse populations. This study highlights the need for more individualized approaches to breast cancer screening and treatment to improve patient outcomes.

**Abstract:**

Background: Differences in the incidence of breast cancer subtypes among racial/ethnic groups have been evaluated as a contributing factor in disparities seen in breast cancer prognosis. We evaluated new breast cancer cases in Hawai’i to determine if there were subtype differences according to race/ethnicity that may contribute to known disparities. Methods: We reviewed 4591 cases of women diagnosed with breast cancer from two large tumor registries between 2015 and 2022. We evaluated breast cancer cases according to age at diagnosis, self-reported race, breast cancer subtype (ER, PR, and HER2 receptor status), histology, county, and year. Results: We found both premenopausal and postmenopausal Native Hawaiian women were less likely to be diagnosed with triple-negative breast cancer (OR = 0.26, 95% CI 0.12–0.58 *p* = 0.001; OR = 0.54, 95% CI 0.36, 0.80 *p* = 0.002, respectively). Conclusions: The results of our study support that there are racial/ethnic differences in breast cancer subtypes among our population, which may contribute to differences in outcomes. Further evaluation of clinical and pathological features in each breast cancer subtype may help improve the understanding of outcome disparities seen among different racial/ethnic groups.

## 1. Introduction

Extensive research has been carried out evaluating the heterogeneous nature of breast cancer according to subtypes and its variable impact among different populations [1,2,3,4]. The prognostic importance of this heterogeneity is complicated by a wide array of underlying behavioral, environmental, social, economic, and biological factors [5,6,7,8,9]. Race/ethnicity is a correlative factor with differences in risk and clinical outcomes of specific breast cancer subtypes. The Carolina Breast Cancer study showed that the triple-negative breast cancer (TNBC) subtype (ER-/PR-/HER2−) was significantly more prevalent, and the luminal A subtype was considerably less prevalent among premenopausal Black women compared to postmenopausal Black women and White women of all ages [1]. These findings suggest that a higher prevalence of triple-negative and lower prevalence of luminal A tumors among premenopausal Black women could explain the poor prognosis and higher mortality rates observed [1,2,5,10]. The results of these studies highlight differences in tumor biology, which may contribute to mortality disparities seen between Black and White women.

There are known racial/ethnic disparities between different groups in Hawai’i’s population. The ethnic population of Hawai’i is diverse, with no racial/ethnic majority group based on population size. About half of the population is of Asian heritage (Japanese, Filipino, Korean, Chinese), about a quarter is of European ancestry (White), and about 20% is Native Hawaiian [11]. Cancer is the leading cause of death in the Asian population, and breast cancer is the most commonly diagnosed cancer among Asian women [12]. Examining the Hawai’i population provides crucial insights into breast cancer subtypes in a population comprised of a high proportion of Asian women [11].

Loo et al. examined differences in incidence and mortality among women who were diagnosed with invasive breast cancer between 1984 and 2013 from the five significant racial/ethnic populations in Hawai’i: Native Hawaiian, non-Hispanic White, Japanese, Chinese, and Filipino. Mortality rates for most individual Asian subpopulations (i.e., Japanese, Chinese, Filipino) were lower compared to non-Hispanic White women; however, Native Hawaiians had the highest mortality rates overall [3]. Native Hawaiians consistently had the highest incidence and mortality rates compared to all racial/ethnic groups in Hawai’i. They were disproportionately affected by poorer survival for both localized and advanced stages at the time of diagnosis [3]. The most common breast cancer subtypes among all patients between 2010 and 2013 were hormone receptor-positive (HR+), followed by triple-negative and human-epidermal growth factor-positive (HER2+) subtypes [3]. Japanese and Native Hawaiians had higher incidence rates of HR+ breast cancer compared to non-Hispanic Whites. Japanese women had higher incidence rates of TNBC compared to non-Hispanic Whites.

Ihenacho et al. characterized breast cancer incidence among Asian American, Native Hawaiian, and non-Hispanic White women in Hawai’i diagnosed with breast cancer between 1990 and 2014. Annual breast cancer incidence increased by 2.9% in premenopausal Japanese and non-Hispanic White women [13]. Among premenopausal women between 2010 and 2014, the incidence was highest in Japanese women [13]. Additionally, the incidence of HR+/HER2− breast cancer was highest in premenopausal Japanese women [13]. Annual breast cancer incidence increased by 1.6% in postmenopausal Filipino women and 4.2% in postmenopausal Japanese women [13]. Among postmenopausal women, breast cancer incidence was highest in Native Hawaiian women, who displayed the highest incidence of HR+/HER- and triple-positive breast cancer [13]. Kong et al. described differences in incidence and distribution among women who were diagnosed with primary unilateral breast cancer who underwent surgical treatment between 2010 and 2015 in the United States and were categorized into five racial groups: American Indian/Alaska Native (AIAN), Hispanic White, Asian American/Pacific Islander (AAPI), Black, and non-Hispanic White. They reported that AAPIs had a higher incidence of the HR−/HER2+ subtype but a lower incidence of HR+/HER2− and TNBC subtypes than non-Hispanic Whites (Kong et al., 2020) [14]. Giaquinto et al. investigated breast cancer trends in the United States between 1995 and 2022. Between 2015 and 2019, the incidence of HR+/HER2− breast cancer was highest in Whites, followed by AAPIs, AIANs, and Blacks [15]. The incidence of HER2+ breast cancer was similar across racial groups [15].

Fong et al. highlighted significant inter- and intra-ethnic variations in female breast cancer incidence in the continental United States and Hawai’i between 1992 and 2002. They found that Asian or Pacific Islanders (API), particularly in Hawai’i, had a more favorable distribution of subtypes, with higher rates of hormone receptor-positive tumors, which are generally associated with better outcomes [16]. The study also reported that APIs in Hawai’i had lower rates of high-risk subtypes compared to their counterparts in the continental U.S., where APIs were predominantly first-generation migrants (92%) [16]. The study demonstrated that API women in Hawai’i exhibited significant heterogeneity within their own group due to the inclusion of Pacific Islanders, who made up a larger portion of the API population in Hawai’i (23.46%) compared to the continental U.S. (2.4%) [16]. This complexity within racial groups underscores the need for disaggregated data, as aggregated racial categories may obscure important differences in breast cancer subtype and prognosis between ethnic subgroups [16].

Prior studies have utilized aggregated AAPI data to discuss breast cancer subtype differences. We have a unique opportunity to study the differences in breast cancer subtypes within these populations individually. We hypothesize that there are breast cancer subtype differences according to race/ethnicity in our population that can contribute to known disparities seen in Hawai’i. We evaluated new breast cancer diagnoses from two major health systems in Hawai’i from 2015 to 2022. We found differences in breast cancer subtypes among racial/ethnic groups in Hawai’i’s uniquely diverse population.

## 2. Materials and Methods

Institutional review board (IRB) approval was obtained from two health systems to evaluate breast cancer subtypes according to race/ethnicity and other demographic factors. Data was collected from the Hawai’i Pacific Health Tumor Registry of women who had an initial date of diagnosis of breast cancer between 1 January 2015 and 31 December 2022, as well as the Queen’s Tumor Registry of women who had an initial date of diagnosis between 1 January 2015 and 31 August 2020. We evaluated the patients’ age at diagnosis, race, breast cancer subtype according to ER, PR, and HER2 receptor status, histology, county, and year.

Race was self-reported and categorized according to five racial/ethnic groups: White, Asian, Filipino, Native Hawaiian or Pacific Islander (NHPI), and Other. The racial/ethnic category designated as Other encompassed women who did not fit into other groups, such as Black and American Indian/Alaskan Native, and whose numbers were insufficient to make a stand-alone group. The first race/ethnicity listed in the self-reported data was used for women with multiple ethnicities. The county was also self-reported and characterized by the location of the patients’ residence at the time of diagnosis. Counties included Honolulu County, Hawai’i County, Maui County, Kauai County, and unknown. Age at diagnosis was stratified into five groups: 18–39 years, 40–49 years, 50–59 years, 60–69 years, and over 70 years.

Tumor subtype and histology were collected from the data reported in tumor registries. Tumor subtype was categorized based on estrogen receptor (ER), progesterone receptor (PR), and human-epidermal growth factor (HER2) status. Categories included triple positive (ER+, PR+, and HER2+), HR+HER2− (ER+ or PR+ and HER2−), HR−HER2+ (ER-, PR-, and HER2+), and triple negative (ER-, PR-, and HER2−). Tumor histology categories included ductal, lobular, mucinous, and other. The other histology category represented other breast tumor types, including angiosarcoma and lymphomas, and, given the nature of their disease, would not be defined by traditional ER/PR/HER2 subtyping.

METRIQ software versions 3.20 (Queen’s Tumor Registry) and 3.50.051.7 (Hawai’i Pacific Health Tumor Registry) were used for data collection and management. The SAS 9.4 (SAS Institute Inc., Cary, NC, USA) software performed all analyses. The LOGISTIC procedure applied multinomial logistic regression analyses on the four-category (triple positive, HR+ HER2−, HR− HER2+, triple negative) subtype outcome variable. Separate models were run for pre- (under or at age 50) and postmenopausal (over age 50) patients. For premenopausal women, the predictors were age at diagnosis, race, and year of diagnosis (entered as a categorical variable). For postmenopausal women, the predictors were age at diagnosis, race, histology, county, and year of diagnosis (entered as a categorical variable). County and histology had insufficient sample size to be included in the premenopausal analysis. All predictor variables were entered simultaneously.

## 3. Results

A total of 4591 cases were evaluated. Of those cases, 902 (19.6%) were age 50 or younger (premenopausal), and 3689 (80.4%) were over the age of 50 (postmenopausal). Chi-square tests assessed overall differences by menopausal status (Table 1). The highest proportion of patients came from Honolulu County (48.2%), followed by an unknown county (39.1%), Kauai County (7.7%), Maui County (3.0%), and Hawai’i County (2.1%). The most represented ethnicity was Asian, with a total of 1799 cases (39.2%), followed by White with 979 cases (21.3%), NHPI with 909 (19.8%), Filipino with 815 cases (17.8%), and other with 89 cases (1.9%) (Table 1). The most common subtype was hormone-positive (ER+ or PR+/HER2−) with a total of 3746 cases (81.6%), followed by triple-negative (ER-/PR-/HER2−) with 415 cases (9.0%), triple-positive (ER+/PR+/HER2+) with 242 cases (5.3%), and HER2+ (ER-/PR-/HER2+) with 188 cases (4.1%) (Table 1).

Among all races, the Asian group had the lowest proportion of premenopausal women (17.7%), while the other group had the highest proportion of premenopausal women (23.6%). Among all breast cancer subtypes, the triple-positive receptor group had the highest proportion of premenopausal women (30.6%), while the HR+HER2− group had the lowest proportion of premenopausal women (18.2%) (Table 1).

Using a multinomial logistic regression analysis, we found statistically significant differences in subtypes for premenopausal women who were Native Hawaiian and Pacific Islander (NHPI), as well as other. Premenopausal NHPI women were 74% less likely to be diagnosed with TNBC (OR = 0.26, 95% CI 0.12, 0.58, *p* = 0.001) compared to White women (Table 2). Additionally, there were statistically significant differences in subtypes according to the year. In 2021, premenopausal women were 4.3 times as likely to be diagnosed with triple-positive breast cancer (OR 4.32, 95% CI 1.65, 11.31, *p* = 0.003) compared to those in 2015. In 2022, premenopausal women were 2.9 times as likely to be diagnosed with TNBC (OR 2.94, 95% CI 1.23, 7.02, *p* = 0.02) compared to those in 2015 (Table 2).

Premenopausal women who were Asian (OR = 0.74, 95% CI 0.43, 1.28, *p* = 0.28), Filipino (OR = 0.57, 95% CI 0.29, 1.14, *p* = 0.11), or other (OR = 1.11, 0.29, 4.27, *p* = 0.88) race had no significant differences in TNBC prevalence compared to premenopausal White women. Asian women tended to have lower rates of triple-positive receptor breast cancer, although this was not statistically significant (OR = 0.49, 95% CI 0.23, 1.02, *p* = 0.06). There were no differences in the prevalence of triple-positive and HR−HER+ breast cancer between all racial groups (Table 2).

In postmenopausal women, the diagnosis of HR−/HER2+ breast cancer was more likely in 2017 (OR = 2.24, 95% CI 1.08, 4.63, *p* = 0.03) and 2018 (OR = 2.91, 95% CI 1.45, 5.85, *p* = 0.003) compared to 2015 (Table 3). Postmenopausal women in the 60–69 years (OR = 0.60, 95% CI 0.41, 0.87, *p* = 0.006) and 70+ years (OR = 0.41, 95% CI 0.27, 0.62, *p* < 0.0001) age groups were less likely to have triple-positive breast cancer compared to women in the 50–59 years age group. NHPI women were less likely to be diagnosed with TNBC compared to White women (OR = 0.54, 95% CI 0.36, 0.80, *p* = 0.002). Postmenopausal women with triple-positive breast cancer were less likely to have lobular histology (OR = 0.17, 95% CI 0.06, 0.46, *p* = 0.0005) compared to ductal histology. Similarly, postmenopausal women with HR−/HER2+ breast cancer were less likely to have lobular histology compared to ductal histology (OR = 0.17, 95% CI 0.05, 0.54, *p* = 0.003). Additionally, postmenopausal women with TNBC were less likely to have lobular (OR = 0.33, 95% CI 0.19, 0.57, *p* < 0.0001) or mucinous (OR = 0.08, 95% CI 0.01, 0.53, *p* = 0.009) histology and were more likely to have a histology categorized as other (OR = 3.08, 95% CI 1.92, 4.95, *p* < 0.0001) compared to ductal histology. Postmenopausal women from an unknown county were less likely to be diagnosed with triple-positive (OR = 0.02, 95% CI 0.00, 0.07, *p* < 0.0001), HR−HER+ (OR = 0.46, 95% CI 0.31, 0.70, *p* = 0.0003), and triple-negative (OR = 0.71, 95% CI 0.54, 0.94, *p* = 0.02) breast cancer compared to those in Honolulu (Table 3). Postmenopausal women who were Asian (OR = 0.81, 95% CI 0.59, 1.11, *p* = 0.19) or Filipino (OR = 0.89, 95% CI 0.61, 1.28, *p* = 0.52) had no significant differences in TNBC prevalence compared to postmenopausal White women. Additionally, there were no differences in the prevalence of triple-positive and HR−HER+ breast cancer between all racial groups. Between postmenopausal women aged 50–59 years, 60–69 years, and 70+ years, there were no differences in the prevalence of TNBC (Table 3).

In premenopausal women, the overall incidence of breast cancer increased from 2015 to 2021, with 62 cases in 2015 and 87 cases in 2021 (*p* = 0.04). In premenopausal White women, the incidence was highest in 2017 (N = 21, *p* = 0.03) and 2020 (N = 24, *p* = 0.01). In premenopausal Asian women, incidence increased between 2015 and 2022, with the highest incidence in 2021 (N = 33, *p* = 0.04) (Table 4, Figure 1 and Figure 2).

In postmenopausal women, the overall incidence of breast cancer did not significantly change between 2015 and 2022 but trended upward between 2017 (N = 281) and 2021 (N = 313). In postmenopausal White women, incidence was lowest in 2022 (N = 57, *p* = 0.02). In postmenopausal Asian women, incidence increased until 2021, with the highest incidences in 2019 (N = 118, *p* = 0.04) and 2021 (N = 117, *p* = 0.04). In postmenopausal NHPI women, the incidence was highest in 2021 (N = 67, *p* = 0.01) (Table 4, Figure 1 and Figure 2).

## 4. Discussion

We report that there are racial and ethnic differences in breast cancer subtypes among Hawai’i’s population. Our data show that both premenopausal and postmenopausal NHPI women are more likely to have hormone-positive breast cancer, a subtype known to have improved outcomes compared to TNBC. Loo et al. similarly found that Native Hawaiian (NH) women had a significantly higher incidence of hormone-positive, triple-positive, and HER2+ breast cancer subtypes compared to White women but a substantially lower risk for the TNBC subtype, which is consistent with the results of our study [3]. It is well established that there are racial/ethnic disparities in breast cancer outcomes in Hawai’i. The Native Hawaiian population experiences the poorest survival for both localized and advanced-stage breast cancer of all racial/ethnic groups in Hawai’i, and Loo et al. suggested that these poor outcomes may be due to biological factors [3]. Furthermore, Taparra et al. reported that NH women diagnosed with ductal carcinoma in situ (DCIS) were more likely to subsequently develop inflammatory breast cancer, a type of breast cancer known to have poor outcomes [17]. These findings suggest that the poor outcomes that Native Hawaiian women with breast cancer experience may be due to factors other than subtype.

Conroy et al. conducted a multiethnic cohort study to examine the differential impact of obesity as a comorbidity on breast cancer survival. They showed that NH women with invasive breast cancer were more likely to have comorbidities such as obesity and cardiovascular disease, pulmonary disease, liver disease, neuromuscular/skeletal disorders, and kidney disease compared to White, Japanese American, and Latino women. Compared to all other ethnic groups studied, obese NH women had a higher risk for all-cause mortality but a lower risk for breast cancer-specific mortality [8]. These findings suggest an interaction between pre-existing comorbid conditions and breast cancer outcomes. The increased prevalence of comorbid conditions in minority populations may contribute to the increased overall mortality rates observed despite better breast cancer-specific prognoses [18]. Another study conducted by Maskarinec et al. showed an inverse relationship between breast cancer-specific mortality and type 2 diabetes mellitus among NH women with invasive breast cancer [9]. They suggest that this may be because NH women are more likely to have regular health visits for comorbidities and, thus, more likely to have early screening for breast cancer [9]. Nevertheless, these collective findings and the results of our study suggest that outcomes of NH women with breast cancer are multifactorial.

Our findings show that premenopausal Japanese women are less likely to have the triple-positive breast cancer subtype compared to White women, which is consistent with findings in California-based studies that Japanese women were found to have lower rates of triple positive and TNBC [19,20,21]. In Hawai’i, postmenopausal Japanese women were found to have higher incidence rates of HR+ breast cancer compared to White women. Still, Japanese women have lower 5-year mortality rates than White and NH women [4]. Japanese women in Hawai’i were also found to have a sharp increase in the incidence of breast cancer, which exceeds that of Whites. Still, Japanese women have maintained relatively low mortality rates [4]. Improved clinical outcomes within this population may be directly attributed to a higher incidence of the molecular subtypes associated with better outcomes, which may also be associated with higher rates for a localized stage at diagnosis and smaller mean tumor size compared to other racial and ethnic groups [4,7,22]. Diagnosis at less advanced stages of the disease may be due to non-biological factors, including access to and frequency of screening, and a lower prevalence of comorbidities, which may play a role in clinical outcomes within this patient population. Despite improved outcomes, previous literature has reported that AAPI women have the lowest rates of up-to-date breast cancer mammography screening compared to other ethnic groups in the United States [23,24]. Disaggregated data demonstrated wide screening rates within the Asian American population, with 93.8% of Japanese compared to 63.3% of Korean women [25]. These findings highlight the need for further evaluation of these groups using disaggregated data to identify specific at-risk populations.

Conversely, late-stage diagnosis may contribute to poorer clinical outcomes in Native Hawaiian, Pacific Islander, and Filipino women. Several studies have reported higher rates of late-stage diagnoses amongst these ethnic groups, which may be explained by barriers to screening and care [26,27,28,29]. Native Hawaiian, Pacific Islander, and Filipino women may not receive recommended cancer screenings due to personal, cultural, practical, knowledge-related, priority-related, and test-related barriers. Personal or cultural barriers include perceptions that the test is too invasive or frightening, the emotional nature, and privacy concerns. Practical barriers include cost and access to care. Test-related barriers involve the time required for preparation or for completing the test. Knowledge-related barriers, such as reading skills, are linked to health literacy [30,31].

Additionally, previous instances in the literature have shown increased periods between diagnosis and intervention, which may further contribute to outcome disparities [7,32,33]. Improved screening programs, education campaigns, and culturally competent care in Hawai’i and similar regions may contribute to mitigating breast cancer disparities in diverse populations. Targeting health interventions specifically for Native Hawaiian and Filipino women could help address some of the identified factors that may contribute to poor breast cancer prognosis. For example, providing pamphlets detailing screening tests in various languages, such as Tagalog, could benefit the community. Additionally, establishing satellite clinics would improve access to healthcare for patients in rural areas.

Postmenopausal Filipino women were significantly more likely to have HER2+ breast cancer compared to Whites. This finding is consistent with California-based studies [19,20,21]. Loo et al. also found that Filipino women with HER2+ breast cancer have poor 5-year survival rates compared to other racial/ethnic groups in Hawai’i [3]. These collective findings suggest that subtype may be a driving factor for clinical outcomes among postmenopausal Filipino women, as these patients tend to have a poor prognosis, as measured by five-year survival for invasive breast cancer, compared to Japanese and White women [3].

Asian women with breast cancer had the lowest proportion of premenopausal cases amongst all racial groups. Amongst all breast cancer subtypes, women in the triple-positive breast cancer subtype group had the highest proportion of premenopausal women. In contrast, women with hormone-receptor-positive breast cancer had the lowest proportion of premenopausal women. Menopause status at diagnosis is known to affect breast cancer prognosis, with premenopausal women often presenting with more aggressive subtypes, while postmenopausal women more commonly have hormone receptor-positive subtypes [1]. Since different racial and ethnic groups may have varying average ages at menopause, this could influence the prevalence of breast cancer subtypes and, thus, clinical outcomes [34]. The variability in age at menopause could contribute to the disparities in breast cancer subtypes observed among Hawai’i’s diverse population. These findings underscore the importance of considering menopause status in the analysis of breast cancer incidence and prognosis, as it may skew the understanding of risk and subtype prevalence across subpopulations.

Our study is the latest to comprehensively analyze breast cancer subtypes among the diverse racial and ethnic groups in Hawai’i. This population-based analysis provides valuable insights into the distribution of breast cancer subtypes and their association with clinical outcomes, highlighting significant disparities. The large sample size and inclusion of multiple racial and ethnic groups add robustness to our findings, making this a significant contribution to understanding breast cancer epidemiology in Hawai’i.

This study has several limitations. We recognize this is not a complete representation of our state based on the Hawai’i Tumor Registry, which reports an annual average number of new breast cancer cases of ~1200 (Hawai’i Tumor Registry-HTR, 2012–2016). The use of tumor registry data also limits the detail of clinical information collected. Furthermore, as this study only included breast cancer cases in Hawai’i, the results may have limited generalizability to other diverse populations with different racial/ethnic compositions. We also acknowledge that factors such as adjuvant treatment and detailed pathological features were not captured, which may introduce bias or limit generalizability. In addition, other pathologic features may contribute to clinical outcomes beyond subtypes, including stage, grade, and adjuvant treatment, which have not been captured in this data review. We also recognize that using self-reported race may lead to misclassification, particularly for individuals of mixed ethnicity. Additionally, racial groups such as Black and AIAN were combined into an “Other” category, and Japanese, Chinese, and other Asians were combined into an “Asian” category to increase sample sizes. This aggregated grouping may obscure meaningful subgroup differences. Additionally, differences in data collection periods between registries could introduce bias or inconsistencies in the results.

Our findings have important implications for clinical practice in Hawai’i and other regions with diverse populations. The identification of specific breast cancer subtypes more prevalent in certain racial and ethnic groups can guide personalized treatment approaches and improve patient outcomes. For example, the higher prevalence of hormone-positive subtypes among NHPI women suggests that tailored endocrine treatments could be particularly beneficial for this group. Additionally, the lower incidence of TNBC among NHPI women indicates a potentially lower risk of aggressive breast cancer forms in this population. Previous studies have focused mainly on continental U.S. populations, leaving a significant gap in knowledge regarding breast cancer epidemiology in Hawai’i. By filling this gap, our study contributes to a more comprehensive understanding of breast cancer disparities and supports the development of targeted interventions to reduce these disparities. The findings from this study underscore the need for tailored breast cancer screening and treatment programs in Hawai’i. Healthcare providers should consider the unique subtype distributions and associated risk factors when developing management plans for breast cancer patients. Policymakers and public health officials should also take these findings into account when designing breast cancer prevention and control programs, ensuring that they address the specific needs of Hawai’i’s diverse population.

While our study provides valuable insights, it raises several questions that warrant further investigation. For example, the reasons behind the poorer outcomes among NHPI women despite a higher prevalence of less aggressive subtypes remain unclear. We recommend conducting longitudinal studies to track breast cancer outcomes, further exploring disparities in healthcare access, and incorporating lifestyle interventions. Because differences in gene expression signatures have been reported in Black patients with breast cancer [35], we will need to define the gene signatures in NHPI and Filipino women, which may contribute to a better understanding of biological differences in breast cancer amongst these populations. Further exploration of advanced therapeutic methods may aid in providing more personalized breast cancer care. Shadbad et al. discussed the potential of Programmed Death-Ligand 1 (PD-L1)-inhibiting microRNAs, delivered via biomimetic carriers guided by single-cell sequencing, to target triple-negative breast cancer by downregulating PD-L1, enhancing immune responses, and inhibiting tumor growth [36]. Their findings are an example of a promising approach for personalized breast cancer therapy, which could help address racial and ethnic disparities in breast cancer outcomes by tailoring treatments based on unique molecular and genetic profiles, enhancing treatment effectiveness, and reducing disparities in high-risk populations. Future research should explore potential biological, environmental, and healthcare access factors that may contribute to breast cancer outcome disparities.

## 5. Conclusions

The results of our study support that there are racial/ethnic differences in breast cancer subtypes among Hawai’i’s population, which may contribute to the outcome differences seen. We are evaluating clinical and pathological features in each breast cancer subtype to understand outcome disparities among racial/ethnic groups in Hawai’i. These differences have important implications for clinical practice, public health programs, and future research. By addressing the unique characteristics of Hawai’’s population, we can improve breast cancer outcomes and reduce disparities, ultimately benefiting all breast cancer patients in the state.

## Figures and Tables

**Figure 1 cancers-16-03462-f001:**
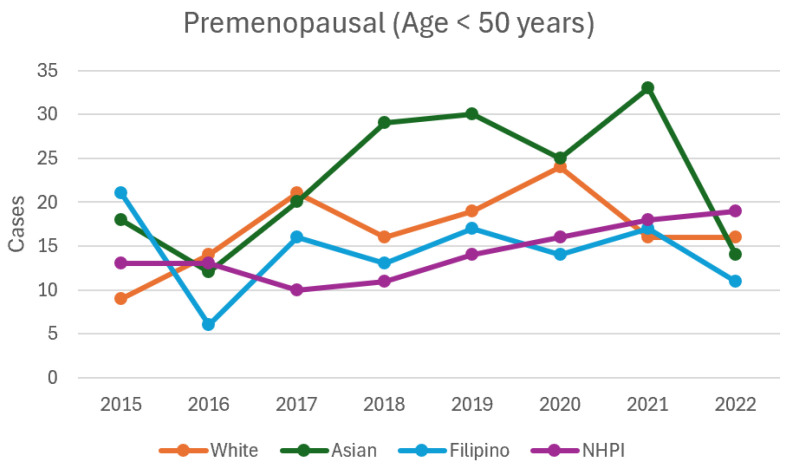
Incidence of breast cancer in premenopausal women from 2015 to 2022.

**Figure 2 cancers-16-03462-f002:**
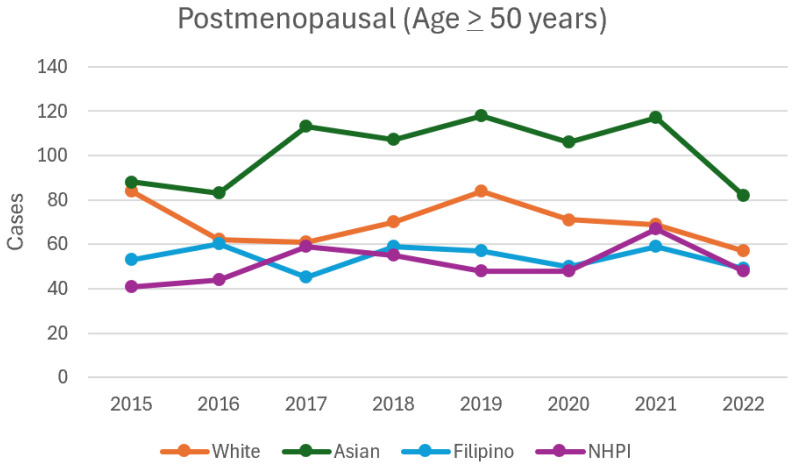
Incidence of breast cancer in postmenopausal women from 2015 to 2022.

**Table 1 cancers-16-03462-t001:** Patient demographics, race, tumor subtype, and tumor histology.

			Total		Premenopausal (Age < 50)		Postmenopausal (Age ≥ 50)	
			N = 4591		N = 902		N = 3689	
Variable		*n*	Col%	*n*	Row%	*n*	Row%	*p*
Age at Diagnosis	18–39	221	4.8	221	100.0	0	0.0	<0.0001
	40–49	681	14.8	681	100.0	0	0.0	
	50–59	993	21.6	0	0.0	993	100.0	
	60–69	1398	30.5	0	0.0	1398	100.0	
	70+	1298	28.3	0	0.0	1298	100.0	
Race	White	979	21.3	190	19.4	789	80.6	0.5
	Asian	1799	39.2	319	17.7	1480	82.3	
	Filipino	815	17.8	179	22.0	636	78.0	
	NHPI	909	19.8	193	21.2	716	78.8	
	other	89	1.9	21	23.6	68	76.4	
Subtype	Triple Positive	242	5.3	74	30.6	168	69.4	<0.0001
	HR+ HER2−	3746	81.6	682	18.2	3064	81.8	
	HR− HER2+	188	4.1	54	28.7	134	71.3	
	Triple Negative	415	9.0	92	22.2	323	77.8	
Histology	Ductal	3873	84.4	790	20.4	3083	79.6	0.15
	Lobular	434	9.5	66	15.2	368	84.8	
	Mucinous	153	3.3	26	17.0	127	83.0	
	Tubular	12	0.3	2	16.7	10	83.3	
	Metaplastic	15	0.3	3	20.0	12	80.0	
	Mixed	23	0.5	4	17.4	19	82.6	
	other	81	1.8	11	13.6	70	86.4	
County	Hawai’i	96	2.1	31	32.3	65	67.7	0.01
	Honolulu	2213	48.2	447	20.2	1766	79.8	
	Kauai	352	7.7	58	16.5	294	83.5	
	Maui	136	3.0	24	17.6	112	82.4	
	unknown	1794	39.1	342	19.1	1452	80.9	
Year	2015	685	14.9	134	19.6	551	80.4	0.53
	2016	672	14.6	117	17.4	555	82.6	
	2017	756	16.5	140	18.5	616	81.5	
	2018	714	15.6	139	19.5	575	80.5	
	2019	628	13.7	128	20.4	500	79.6	
	2020	430	9.4	96	22.3	334	77.7	
	2021	400	8.7	87	21.8	313	78.3	
	2022	306	6.7	61	19.9	245	80.1	

**Table 2 cancers-16-03462-t002:** Differences in breast cancer subtype by year, age, and race in premenopausal women.

			Total				Triple Positive				HR+ HER2−				HR− HER2+						Triple Negative		
		N	Col%	N	Row%	OR	LCL	UCL	*p*	N	Row%	N	Row%	OR	LCL	UCL	*p*	N	Row%	OR	LCL	UCL	*p*
Total		902	100	74	8.2					682	75.6	54	6.0					92	10.2				
Age	18–39	221	24.5	31	14.0	1.00				137	62.0	22	10.0	1.00				31	14.0	1.00			
Age	40–49	681	75.5	43	6.8	0.38	0.23	0.63	0.0002	545	79.7	32	4.7	0.36	0.20	0.65	0.0007	61	8.9	0.49	0.30	0.81	0.005
Race	White	190	21.1	17	8.9	1.00				135	71.1	9	4.7	1.00				29	15.3	1.00			
Race	Asian	319	35.4	16	4.7	0.49	0.23	1.02	0.06	251	77.2	16	5.8	1.13	0.48	2.66	0.79	36	12.3	0.74	0.43	1.28	0.28
Race	Filipino	179	19.8	20	11.2	1.25	0.61	2.54	0.55	130	71.3	14	8.8	1.85	0.76	4.48	0.18	15	8.8	0.57	0.29	1.14	0.11
Race	NHPI	193	21.4	18	8.8	0.88	0.43	1.81	0.73	153	79.7	13	7.0	1.31	0.54	3.21	0.55	9	4.5	0.26	0.12	0.58	0.001
Race	other	21	2.3	3	12.4	1.58	0.39	6.41	0.52	13	62.1	2	10.8	2.60	0.50	13.65	0.26	3	14.8	1.11	0.29	4.27	0.88
Year	2015	134	14.9	7	5.2	1.00				108	80.6	6	4.5	1.00				13	9.7	1.00			
Year	2016	117	13.0	7	6.0	1.14	0.38	3.42	0.81	95	81.3	8	7.0	1.54	0.51	4.67	0.44	7	5.7	0.58	0.22	1.54	0.28
Year	2017	140	15.5	5	3.4	0.64	0.20	2.11	0.47	115	82.5	11	7.9	1.73	0.61	4.89	0.30	9	6.1	0.62	0.25	1.52	0.29
Year	2018	139	15.4	11	8.0	1.50	0.55	4.08	0.42	113	82.0	6	4.1	0.90	0.28	2.93	0.87	9	5.9	0.59	0.24	1.46	0.26
Year	2019	128	14.2	15	10.9	2.34	0.89	6.12	0.08	88	71.9	7	4.9	1.23	0.39	3.88	0.72	18	12.3	1.42	0.65	3.13	0.38
Year	2020	96	10.6	7	6.6	1.40	0.46	4.26	0.55	67	72.8	8	8.0	1.98	0.64	6.09	0.23	14	12.5	1.43	0.62	3.30	0.40
Year	2021	87	9.6	16	18.5	4.32	1.65	11.31	0.003	57	66.2	5	5.6	1.52	0.44	5.27	0.51	9	9.7	1.22	0.49	3.07	0.67
Year	2022	61	6.8	6	9.4	2.29	0.71	7.37	0.16	39	63.3	3	4.9	1.40	0.33	5.96	0.65	13	22.4	2.94	1.23	7.02	0.02

Col% = Column %; OR = odds ratio; LCL = lower confidence limit; UCL = upper confidence limit; NHPI = Native Hawaiian or Pacific Islander.

**Table 3 cancers-16-03462-t003:** Differences in breast cancer subtype by age, race, histology, county, and year in postmenopausal women.

			Total				Triple Positive				HR+ HER2−				HR− HER2+						Triple Negative		
		N	Col%	N	Row%	OR	LCL	UCL	*p*	N	Row%	N	Row%	OR	LCL	UCL	*p*	N	Row%	OR	LCL	UCL	*p*
Total		3689	100	168	4.6					3064	83.1	134	3.6					323	8.8				
Age	50–59	993	26.9	72	7.3	1.00				803	80.9	44	4.4	1.00				74	7.5	1.00			
Age	60–69	1398	37.9	58	4.4	0.60	0.41	0.87	0.006	1170	83.1	49	3.7	0.81	0.53	1.24	0.34	121	8.7	1.14	0.84	1.55	0.41
Age	70+	1298	35.2	38	3.1	0.41	0.27	0.62	<0.0001	1091	84.0	41	3.2	0.70	0.45	1.10	0.12	128	9.7	1.25	0.92	1.71	0.15
Race	White	789	21.4	43	5.4	1.00				641	81.2	23	2.9	1.00				82	10.4	1.00			
Race	Asian	1480	40.1	51	4.1	0.72	0.46	1.14	0.16	1247	84.0	54	3.2	1.07	0.63	1.79	0.81	128	8.7	0.81	0.59	1.11	0.19
Race	Filipino	636	17.2	38	5.6	1.04	0.64	1.68	0.89	504	80.5	36	4.8	1.66	0.95	2.89	0.07	58	9.1	0.89	0.61	1.28	0.52
Race	NHPI	716	19.4	33	4.6	0.80	0.48	1.32	0.37	621	86.9	20	2.5	0.79	0.42	1.48	0.46	42	6.0	0.54	0.36	0.80	0.002
Race	other	68	1.8	3	5.9	1.19	0.33	4.24	0.79	51	73.4	1	1.5	0.56	0.07	4.25	0.57	13	19.3	2.06	1.05	4.03	0.03
Histology	Ductal	3083	83.6	155	5.0	1.00				2519	81.7	127	4.1	1.00				282	9.1	1.00			
Histology	Lobular	368	10.0	4	1.0	0.17	0.06	0.46	0.0005	347	94.7	3	0.8	0.17	0.05	0.54	0.003	14	3.5	0.33	0.19	0.57	0.0001
Histology	Mucinous	127	3.4	3	2.2	0.37	0.11	1.20	0.10	123	97.0	0	0.0	0.00			0.96	1	0.8	0.07	0.01	0.53	0.009
Histology	other	111	3.0	6	4.3	1.02	0.43	2.44	0.96	75	68.6	4	3.5	1.00	0.35	2.81	0.99	26	23.7	3.08	1.92	4.95	<0.0001
County	Hawai’i	65	1.8	8	10.5	1.47	0.66	3.28	0.35	49	77.0	3	5.0	1.12	0.33	3.77	0.86	5	7.5	0.82	0.31	2.12	0.68
County	Honolulu	1766	47.9	129	7.3	1.00				1391	78.8	81	4.6	1.00				165	9.3	1.00			
County	Kauai	294	8.0	15	4.4	0.59	0.33	1.04	0.07	236	81.6	10	3.5	0.74	0.37	1.49	0.40	33	10.4	1.08	0.71	1.64	0.72
County	Maui	112	3.0	14	10.4	1.40	0.75	2.63	0.30	88	80.2	1	0.9	0.19	0.03	1.43	0.11	9	8.5	0.89	0.43	1.83	0.75
County	unknown	1452	39.4	2	0.1	0.02	0.00	0.07	<0.0001	1300	89.9	39	2.4	0.46	0.31	0.70	0.0003	111	7.6	0.71	0.54	0.94	0.02
Year	2015	551	14.9	21	3.8	1.00				475	86.2	11	2.0	1.00				44	8.0	1.00			
Year	2016	555	15.0	11	2.0	0.54	0.25	1.14	0.11	474	84.6	18	3.3	1.69	0.78	3.63	0.18	52	10.1	1.29	0.84	1.98	0.24
Year	2017	616	16.7	30	5.6	1.52	0.84	2.77	0.17	515	82.4	25	4.3	2.24	1.08	4.63	0.03	46	7.8	1.02	0.66	1.58	0.93
Year	2018	575	15.6	13	2.1	0.57	0.28	1.18	0.13	486	84.5	33	5.7	2.91	1.45	5.85	0.003	43	7.7	0.98	0.63	1.53	0.93
Year	2019	500	13.6	23	3.6	1.00	0.53	1.87	0.99	405	81.9	16	3.1	1.61	0.74	3.54	0.23	56	11.4	1.50	0.98	2.30	0.06
Year	2020	334	9.1	28	5.0	1.39	0.75	2.55	0.29	256	81.6	14	3.7	1.95	0.86	4.43	0.11	36	9.7	1.28	0.79	2.07	0.32
Year	2021	313	8.5	19	2.9	0.75	0.39	1.45	0.39	260	87.8	9	2.3	1.11	0.45	2.77	0.82	25	7.0	0.86	0.51	1.48	0.59
Year	2022	245	6.6	23	4.9	1.31	0.69	2.46	0.41	193	84.7	8	2.8	1.40	0.55	3.61	0.48	21	7.6	0.97	0.55	1.71	0.92

Col% = Column %; OR = odds ratio; LCL = lower confidence limit; UCL = upper confidence limit; NHPI = Native Hawaiian or Pacific Islander.

**Table 4 cancers-16-03462-t004:** Incidence of breast cancer by race from 2015 to 2022.

				Premenopausal (Age < 50)				Postmenopausal (Age ≥ 50)	
RACE	Year	Cases	LCL	UCL	*p*	Cases	LCL	UCL	*p*
All	2015	62	48.3	79.5		271	240.6	305.3	
All	2016	46	34.5	61.4	0.13	253	223.7	286.2	0.43
All	2017	69	54.5	87.4	0.54	281	250.0	315.9	0.67
All	2018	72	57.2	90.7	0.39	296	264.1	331.7	0.29
All	2019	83	66.9	102.9	0.08	313	280.2	349.7	0.08
All	2020	82	66.0	101.8	0.10	279	248.1	313.7	0.73
All	2021	87	70.5	107.3	0.04	313	280.2	349.7	0.08
All	2022	61	47.5	78.4	0.93	245	216.2	277.7	0.25
White	2015	9	4.7	17.3		84	67.8	104.0	
White	2016	14	8.3	23.6	0.30	62	48.3	79.5	0.07
White	2017	21	13.7	32.2	0.03	61	47.5	78.4	0.06
White	2018	16	9.8	26.1	0.17	70	55.4	88.5	0.26
White	2019	19	12.1	29.8	0.06	84	67.8	104.0	0.99
White	2020	24	16.1	35.8	0.01	71	56.3	89.6	0.30
White	2021	16	9.8	26.1	0.17	69	54.5	87.4	0.23
White	2022	16	9.8	26.1	0.17	57	44.0	73.9	0.02
Asian	2015	18	11.3	28.6		88	71.4	108.4	
Asian	2016	12	6.8	21.1	0.28	83	66.9	102.9	0.70
Asian	2017	20	12.9	31.0	0.75	113	94.0	135.9	0.08
Asian	2018	29	20.2	41.7	0.11	107	88.5	129.3	0.17
Asian	2019	30	21.0	42.9	0.09	118	98.5	141.3	0.04
Asian	2020	25	16.9	37.0	0.29	106	87.6	128.2	0.20
Asian	2021	33	23.5	46.4	0.04	117	97.6	140.2	0.04
Asian	2022	14	8.3	23.6	0.48	82	66.0	101.8	0.65
Filipino	2015	21	13.7	32.2		53	40.5	69.4	
Filipino	2016	6	2.7	13.4	0.007	60	46.6	77.3	0.51
Filipino	2017	16	9.8	26.1	0.41	45	33.6	60.3	0.42
Filipino	2018	13	7.5	22.4	0.17	59	45.7	76.1	0.57
Filipino	2019	17	10.6	27.3	0.52	57	44.0	73.9	0.70
Filipino	2020	14	8.3	23.6	0.24	50	37.9	66.0	0.77
Filipino	2021	17	10.6	27.3	0.52	59	45.7	76.1	0.57
Filipino	2022	11	6.1	19.9	0.08	49	37.0	64.8	0.69
NHPI	2015	13	7.5	22.4		41	30.2	55.7	
NHPI	2016	13	7.5	22.4	0.99	44	32.7	59.1	0.74
NHPI	2017	10	5.4	18.6	0.53	59	45.7	76.1	0.07
NHPI	2018	11	6.1	19.9	0.68	55	42.2	71.6	0.15
NHPI	2019	14	8.3	23.6	0.85	48	36.2	63.7	0.46
NHPI	2020	16	9.8	26.1	0.58	48	36.2	63.7	0.46
NHPI	2021	18	11.3	28.6	0.37	67	52.7	85.1	0.01
NHPI	2022	19	12.1	29.8	0.29	48	36.2	63.7	0.46

OR = odds ratio; LCL = lower confidence limit; UCL = upper confidence limit; NHPI = Native Hawaiian or Pacific Islander.

## Data Availability

The datasets used and/or analyzed during the current study are available from the corresponding author upon reasonable request.

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
