# Peer review of "Differences in Breast Cancer Subtypes among Racial/Ethnic Groups"

_cancers, 2024, doi:10.3390/cancers16203462_

Round 1

Reviewer 1 Report

Comments and Suggestions for Authors

The authors evaluated breast cancer subtypes among the diverse Hawaiian population to assess possible reasons for disparities in outcomes. 

I have the following minor comments:

1.  Introduction:  It would be useful to cite the earlier study by Fong M, Henson DE, Devesa SS, Anderson WF.  Inter-and intra-ethnic differences for female breast carcinoma incidence in the Continental US and in the state of Hawaii.  Breast Cancer Reseach and Treatment (2006) 97:57-65. 

2. Introduction:  It would be helpful to add a statement in the first paragraph regarding the overall significance of cancer as the leading cause of death in the Asian population and breast cancer as the most commonly diagnosed cancer in Asian women. 

2. Methods:  What International Classification of Diseases for Oncology codes were used to identify histopathological subtypes?

3.  Discussion:  Given that menopause status at diagnosis has a complex association with long term outcome, to what extent might variability in average age at menopause by subpopulation impact the results?

4.  Discussion: Page 9 line 216:  It is mentioned that access to and frequency of screening might play a role. It might be useful to discuss in more detail about what is known, and what might be the impact of different mammography rates by subpopulation.

Reviewer 2 Report

Comments and Suggestions for Authors

Thank you for the opportunity to review this interesting report on the Differences in Breast Cancer Subtypes Among Racial/Ethnic Groups. I found the manuscript enjoyable and insightful. However, while the paper has many strengths, there are several areas where greater clarity would be beneficial. I believe the paper could be further improved by providing additional information on the following points:

1.      What does first race/ethnicity mean? And how did you define it?

2.      Could you please provide a clear explanation for why Black and AIAN women were combined into the Other category? This would help in understanding the race/ethnicity definitions used in the study.

3.      Please explain in the method section if any adjustment was included in the multinomial logistic model. If yes, what variables were used for adjustment? And why?  

4.      In the results, the authors mentioned multivariate logistic regression. However, multinomial logistic regression was mentioned in the methods. It is not clear whether multivariate, multivariable, or multinomial logistic regression was used. However, a combination of multivariable and multinomial is also possible. This is very confusing. Please make the methods and results clearer.

5.      In the results, the authors reported statistically significant differences between 2021, 2022, and 2015. However, it is unclear in the methods section how time was added to the model. Please clarify this in the method section.

6.      The method section has six race categories: Chinese, Filipino, Japanese, Native Hawaiian (NH), White, and Other. However, nine race categories were presented in Table 1, including Asian, Black, and other. The authors mentioned that “The racial/ethnic category designated as ‘Other’ encompassed women who did not fit into other groups, such as Black, and AIAN women” in the method section. In addition, the results of only 5 categories were presented in Table 2. The race categories and their inclusion in the descriptive table and logistic regression are unclear. Please revise the method section and make the categories consistent throughout the results.

7.      Authors reported that “premenopausal Japanese women were 63% less likely to be diagnosed with triple-positive breast cancer (OR=0.37, P=0.03) compared to White women (Table 2).” However, there is Japanese group in Table 2.

8.      Age categories are different in Table 2 and Table 3. Please explain the reason for the different age categories in the method section.

9.      The number of variables included in the model is different in Table 2 and Table 3. Please explain why different sets of variables were chosen for each analysis in the method section.

10.  Not all race categories were presented in Table 2. Please explain why in the method section.

11.  How do you interpret the result from the unknown county? Do you have other information about them? What could be the reason for significant differences?

12.  The number of cases for the incidence of BC in 2021 for premenopausal women should be 87, as reported in Table 4. In addition, the number of cases of BC in premenopausal White women in 2020 should be 24. Please double-check all numbers in the table with the text.

13.  Figures need captions explaining the abbreviations and a summary of the important findings from the figures.

Reviewer 3 Report

Comments and Suggestions for Authors

The authors conducted a study focused on breast cancer subtypes among Hawaii’s diverse ethnic population to explore potential disparities in breast cancer outcomes across different racial and ethnic groups. They analyzed 4,591 breast cancer cases from tumor registries between 2013 and 2022, considering factors like age, race, tumor biology, and subtype (ER, PR, HER2). The findings revealed significant racial/ethnic differences in breast cancer subtypes. Native Hawaiian women, both pre- and post-menopausal, were less likely to be diagnosed with triple-negative breast cancer, while premenopausal Japanese women had a lower likelihood of being diagnosed with triple-positive breast cancer. These subtype-specific differences may contribute to observed disparities in outcomes. The study underscores the need for more individualized approaches to breast cancer treatment and screening, especially in regions with ethnically diverse populations, to improve prognosis and reduce mortality rates.

The study had several limitations that could affect the interpretation of the findings:

  1. Self-reported race/ethnicity: The reliance on self-reported race may lead to misclassification, especially for individuals of mixed ethnicity, where only the first ethnicity was considered.

  2. Limited racial/ethnic categories: The study grouped women who did not fit into the predefined racial categories (e.g., Black, American Indian/Alaska Native) into an "Other" category, which may obscure subgroup differences.

  3. Data range discrepancies: The data collection periods differed slightly between the two tumor registries, with one ending in 2020 and the other continuing until 2022. This could introduce inconsistencies in the data.

  4. Focus on a specific population: The study focused solely on women in Hawaii, which may limit the generalizability of the results to other populations with different racial/ethnic compositions.

  5. Menopausal status based on age: Menopausal status was determined by age (under or over 50), which may not accurately represent individual menopausal status and could affect the analysis.

  6. Tumor registry data: The study used tumor registry data, which may lack detailed clinical information or updates on patient outcomes beyond diagnosis.

These limitations suggest that further research with more precise classifications and broader populations is needed to fully understand breast cancer disparities. If beyond the scope of the manuscript, at least highlight these limitations.

To strengthen the discussion section, several improvements can be made to provide a more comprehensive interpretation of the findings and their broader implications:

  1. Expand on Biological and Genetic Contributions: The discussion could benefit from exploring potential biological and genetic factors that may underlie the racial and ethnic disparities in breast cancer subtypes. Future research directions should be mentioned, such as genetic studies that focus on NH and Filipino women, who experience poor outcomes despite having less aggressive subtypes.

  2. Clarify the Role of Comorbidities: The study highlights the high prevalence of comorbidities like obesity and cardiovascular disease in NH women. The discussion could delve deeper into how these comorbidities interact with breast cancer outcomes, potentially increasing overall mortality despite better breast cancer-specific prognoses. This connection warrants further research into integrative care that addresses both cancer and comorbid conditions.

  3. Highlight Gaps in Healthcare Access and Social Determinants of Health: To explain the poor survival rates among NH and Filipino women, the discussion could explore the role of healthcare access, socioeconomic factors, and cultural barriers. Including references to studies that examine healthcare disparities (e.g., access to early screening, timely diagnosis, and high-quality treatment) would help contextualize these outcomes within the broader social determinants of health.

  4. Consider the Impact of Stage at Diagnosis: While subtype differences are significant, the stage at diagnosis is also crucial for survival. The discussion could emphasize the need for future studies to assess whether the disparities in breast cancer outcomes are partly driven by late-stage diagnosis, especially in underserved populations. Identifying how to promote earlier detection in NH and Filipino women should be a key area for future intervention.

  5. Explore Treatment Accessibility and Adherence: The discussion could expand on whether differences in treatment access and adherence contribute to the disparities observed in NH and Filipino women. For instance, exploring whether these groups face barriers to receiving timely or adequate treatment (e.g., systemic therapy, surgery, or radiation) may offer insights into the observed outcomes.

  6. Connect with Broader Public Health Initiatives: The discussion should tie the findings into broader public health efforts to reduce breast cancer disparities in diverse populations. Suggestions for improved screening programs, education campaigns, and culturally competent care in Hawaii and similar regions would strengthen the clinical relevance of the study. For example, targeting health interventions specifically for NH and Filipino women could mitigate some of the risk factors identified.

  7. Future Research Directions: The discussion should clearly outline future research priorities, such as longitudinal studies to track breast cancer outcomes over time, investigations into healthcare access disparities, and the integration of lifestyle interventions to reduce the impact of comorbidities. Additionally, genetic research into breast cancer subtypes prevalent in these ethnic groups could help identify biological contributors to these disparities.

  8. Addressing Limitations: While some limitations were noted, the discussion could acknowledge the potential impact of these limitations in greater detail. For example, explain how the reliance on self-reported race/ethnicity and differences in data collection periods between registries could introduce bias or affect generalizability. Additionally, acknowledge that other critical factors, such as adjuvant treatment and detailed pathological features, were not captured.

By incorporating these points, the discussion section can offer a more nuanced analysis of the findings, propose meaningful public health interventions, and suggest future research directions that address the complexities of breast cancer disparities in Hawaii.

Finally, Advanced therapeutic strategies like PD-L1 inhibitors, microRNA-based gene therapy, biomimetic carriers (such as leukosomes), and single-cell sequencing offer promising avenues for improving personalized medicine in breast cancer, particularly for triple-negative breast cancer (TNBC). These approaches could help address racial and ethnic disparities in breast cancer outcomes by tailoring treatments based on unique molecular and genetic profiles, ultimately enhancing treatment effectiveness and reducing disparities in high-risk populations (refer to PMID: 34440380 and expand the introduction and discussion sections).

Comments on the Quality of English Language

Minor typos.

Round 2

Reviewer 2 Report

Comments and Suggestions for Authors

Dear Authors,

Thank you for addresing my concerns. I believe the new version of the manuscript is suitable for publication.

Reviewer 3 Report

Comments and Suggestions for Authors

The authors have clarified several of the questions I raised in my previous review. Most of the major problems have been addressed by this revision

Comments on the Quality of English Language

Fine.